# Prognostic, Immunological, and Mutational Analysis of MTA2 in Pan-Cancer and Drug Screening for Hepatocellular Carcinoma

**DOI:** 10.3390/biom13060883

**Published:** 2023-05-24

**Authors:** Xueshan Huang, Jingyi Tan, Mei Chen, Weirang Zheng, Shanyang Zou, Xiaoxia Ye, Yutong Li, Minhua Wu

**Affiliations:** 1The First Clinical Medical College, Guangdong Medical University, Zhanjiang 524000, China; xueshan@gdmu.edu.cn (X.H.);; 2School of Pharmacy, Guangdong Medical University, Zhanjiang 524000, China; 3School of Basic Medicine, Guangdong Medical University, Zhanjiang 524000, China

**Keywords:** MTA2, pan-cancer analysis, immune cell infiltration, MK-886, immunohistochemistry

## Abstract

Background: Metastasis-associated protein 2 (MTA2) is a member of the metastasis-associated transcriptional regulator family and is a core component of the nucleosome remodeling and histone deacetylation complex. Despite growing evidence that MTA2 plays a crucial role in the tumorigenesis of certain cancers, no systematic pan-cancer analysis of MTA2 is available to date. Therefore, the aim of our study is to explore the prognostic value of MTA2 in 33 cancer types and to investigate its potential immune function. Methods: by comprehensive use of databases from TCGA, GTEx, GEO, UCSC xena, cBioPortal, comPPI, GeneMANIA, TCIA, MSigDB, and PDB, we applied various bioinformatics approaches to investigate the potential role of MTA2, including analyzing the association of MTA2 with MSI, prognosis, gene mutation, and immune cell infiltration in different tumors. We constructed a nomogram in TCGA-LIHC, performed single-cell sequencing (scRNA-seq) analysis of MTA2 in hepatocellular carcinoma (HCC), and screened drugs for the treatment of HCC. Finally, immunohistochemical experiments were performed to verify the expression and prognostic value of MTA2 in HCC. In vitro experiments were employed to observe the growth inhibition effects of MK-886 on the HCC cell line HepG2. Results: The results suggested that MTA2 was highly expressed in most cancers, and MTA2 expression was associated with the prognosis of different cancers. In addition, MTA2 expression was associated with Tumor Mutation Burden (TMB) in 12 cancer types and MSI in 8 cancer types. Immunoassays indicated that MTA2 positively correlated with activated memory CD4 T cells and M0 macrophage infiltration levels in HCC. ScRNA-seq analysis based on the GEO dataset discovered that MTA2 was significantly expressed in T cells in HCC. Finally, the eXtreme Sum (Xsum) algorithm was used to screen the antitumor drug MK-886, and the molecular docking technique was utilized to reveal the binding capacity between MK-886 and the MTA2 protein. The results demonstrated excellent binding sites between them, which bind to each other through Π-alkyl and alkyl interaction forces. An immunohistochemistry experiment showed that MTA2 protein was highly expressed in HCC, and high MTA2 expression was associated with poor survival in HCC patients. MK-886 significantly inhibited the proliferation and induced cell death of HepG2 cells in a dose-dependent manner. Conclusions: Our study demonstrated that MTA2 plays crucial roles in tumor progression and tumor immunity, and it could be used as a prognostic marker for various malignancies. MK-886 might be a powerful drug for HCC.

## 1. Introduction

Cancer is one of the leading causes of death worldwide. Lung, liver, stomach, breast, and colon cancers are the top five leading causes of cancer death today, and liver cancer grew from the third-highest cancer death rate in 2018 to the second-highest in 2020 [1]. Hepatocellular carcinoma (HCC) accounts for approximately 85% of liver cancers. In contrast to the decline in death rates for other common cancers, death from liver cancer continues to increase approximately 2–3% annually because liver cancer is often diagnosed at a late stage and there are no curative treatments for advanced liver cancer [2]. Data mining techniques have been at the forefront of medical research and have been proven to be excellent in assessing patient risk and assisting clinical decisions in building disease prediction models [3]. For this reason, data mining has specific advantages in clinical big data research, especially in large-scale medical public databases.

In recent years, immunotherapy has become an effective therapeutic strategy for cancer, especially immune checkpoint blockade therapy [4]. Immune checkpoint inhibitors (ICIs) have the capability to stimulate T cell activation and proliferation, enabling them to destroy tumor cells more effectively [5]. The concept of Tumor Mutation Burden (TMB) is defined as the total number of mutations present in the tumor tissues. Higher TMB results in more neoantigens and increased T cell recognition and is associated with better outcomes of ICIs-based therapy [6]. The tumor microenvironment (TME) is where many of the non-cancerous cells affect the survival and activity of cancer cells [7]. Therapeutic strategies targeting the TME have emerged as a promising approach to cancer treatment due to the critical role of the tumor microenvironment in regulating tumor progression and response to standard tumor therapy [8]. Oncogene-derived changes in tumor cell metabolism can influence the TME to limit the immune response, thus hindering cancer therapy [9]. The TME can deplete some of the nutrients and keep cancer cells proliferating through nutrient removal mechanisms, allowing cancer cells to adapt [10]. ICIs targeting the cytotoxic T lymphocyte-associated protein 4 (CTLA-4) and programmed cell death protein-1 (PD-1)/programmed cell death ligand-1 (PD-L1) pathways are effective treatments for advanced malignancies [11]. Blockade of CTLA-4 leads to activation and an increase in proliferating T cells. PD-1 is expressed in T lymphocytes and other immune cells, while PD-L1 is one of its ligands. The combination of PD-1 and PD-L1 induces suppressive signals in T cells [12,13,14]. Although ICIs such as anti-CTLA-4, anti-PD-1, and anti-PD-L1 have been developed, their overall therapy efficiency for liver cancer is low [15]. Therefore, the development of new target drugs is urgent for HCC treatment.

Cancer metastasis is the process of cancer cells spreading from the primary lesion to distant organs and is the dominant cause of cancer death [16]. MTA2, a member of the metastatic tumor-associated transcriptional regulator family, is a core component of the nucleosome remodeling and histone deacetylation complex [17]. MTA2 is upregulated in human cancers and is associated with an aggressive phenotype, treatment resistance, and a poor prognosis in cancer patients [18]. MTA2 is amplified in a variety of cancers, including HCC [19], thyroid cancer, and ovarian cancer [20]. MTA2 promotes the growth of pancreatic ductal adenocarcinoma cells via inhibition of phosphatase and tensin homolog (PTEN) [21]. MTA2 knockdown inhibits the invasion and metastasis of kidney cancer cells in vivo with no effect on cell proliferation [22]. MTA2 expression significantly promotes the growth, metastasis, and epithelial mesenchymal transition (EMT) progression of esophageal squamous carcinoma [23]. MTA2 expression is regulated by the transcription factor Sp1 in gastric cancer and is capable of promoting the migration and invasion of gastric cancer cells [24]. MTA2 promotes HCC progression mostly by regulating the p38MAPK/MMP2 [25] and Hippo/FRMD6 [20] pathways. In conclusion, increasing evidence demonstrates that MTA2 is an important regulator of tumor progression and that its expression can affect the prognosis of cancer patients. However, so far, most studies on MTA2 are limited to a specific type of cancer, and there is no systematic analysis of MTA2 pan-cancer. Therefore, it is necessary to conduct a pan-cancer analysis of MTA2.

In this study, we applied several databases for combined analysis, followed by validation with an immunohistochemistry experiment. The TCGA [26] and cBioPortal [27] databases were used to explore the relationship between MTA2 expression levels and the prognosis of different types of malignancies. The GEO database [28] was applied to analyze the extent of T cell infiltration at the single cell level of MTA2 in HCC. The MSigDB [29] database was used to analyze the MTA2 and pan-cancer immune linkages. The PubChem database [30] was utilized to screen potent drugs against HCC. Moreover, we conducted molecular docking to explore the binding capacity of the MTA2 protein and drug. We also explored the association of MTA2 expression with MSI and TMB in 33 cancer types. Finally, we investigated the biological function of MTA2 in cancer. The results suggested that MTA2 can act as a prognostic biomarker in a variety of cancers and that MTA2 plays an essential role in tumor immunity by affecting tumor-infiltrating immune cells, TMB, and MSI. We also explored immune function and drug prediction based on MTA2 expression in HCC. Immunohistochemistry was performed to explore MTA2 expression and its association with the survival of HCC patients. This study can provide insight into the role of MTA2 in tumor immunotherapy.

## 2. Materials and Methods

### 2.1. Pan-Cancer Data Acquisition

Applying the UCSC database (University of California Santa Cruz, https://xena.ucsc.edu/, accessed on 14 July 2022), we obtained RNA sequencing, somatic mutations, and relevant clinical data (containing 11,057 samples of 33 cancers) from the Cancer Genome Atlas database (TCGA, https://portal.gdc.cancer.gov/, accessed on 14 July 2022). We downloaded gene expression data from the Genotype-Tissue Expression database for normal tissues (GTEx, https://commonfund.nih.gov/GTEx, accessed on 14 July 2022). Single-cell sequencing data for HCC were acquired from the Gene Expression Omnibus database for analysis (GEO, https://www.ncbi.nlm.nih.gov/gds, accessed on 26 August 2022).

### 2.2. Construction of PPI Networks of MTA2

The protein–protein interaction (PPI) network and significant signaling pathways of MTA2 were investigated using the GeneMANIA (http://genemania.org/, accessed on 24 September 2022) and comPPI (Compartmentalized Protein–Protein Interaction Database, v2.1.1, https://comppi.linkgroup.hu/, accessed on 24 September 2022) databases. We performed network and pathway analysis by applying the GeneMANIA gene network tool, which contains 353 human interaction networks, on the basis of data from BIND, IntAct, and other interaction databases. Association data from protein and genetic interactions, known and predicted pathways, co-expression, co-localization, and protein structural domain similarities were utilized [31]. The comPPI database was applied to identify biologically meaningful and high-confidence interactions between proteins with similar subcellular localization patterns. The comPPI is a protein subcellular localization and protein–protein interaction network from multiple databases. High-confidence interaction sets for each member of the Homo sapiens bispecific phosphatase family were obtained from comPPI filtered with a threshold of 0.7 for each of the localization and interaction scores from a previous study [32].

### 2.3. The Mutational Landscape of MTA2 in Pan-Cancer

The cBioPortal database (https://www.cbioportal.org, accessed on 27 September 2022) was employed to analyze the copy number alteration of MTA2 in pan-cancer. We also analyzed the mutational landscape of MTA2 in pan-cancer from the SangerBox online website (http://sangerbox.com/, accessed on 29 September 2022). We investigated the loci of MTA2 on human chromosomes. We analyzed the mutation of MTA2 and its co-expressed genes in pan-cancer from the UCSC database.

### 2.4. Association of MTA2 with Prognosis in Pan-Cancer

We acquired survival data from TCGA for 33 cancer types. The overall survival (OS), disease-free survival (DFS), disease-specific survival (DSS), and progression-free survival (PFS) were applied to investigate the relationship between MTA2 and the prognosis of tumor patients. The Kaplan-Meier method and log-rank test were employed for survival analysis by cancer type (*p* < 0.05). Survival curves were constructed utilizing the R packages “survival” and “survminer”. Cox analysis was performed using the R packages “survival” and “forestplot” to determine the relationship between MTA2 expression and the survival of pan-cancer patients.

### 2.5. Correlation of MTA2 Expression with Tumor Mutation Burden (TMB) and Microsatellite Instability (MSI)

TMB reflects the number of cancer mutations. Mutations are processed as neoantigens and presented to T cells by major histocompatibility complex (MHC) proteins [6]. MSI has also been observable in approximately 15% of sporadic colorectal cancer (CRC), gastric cancer (GC), and endometrial cancer (EC), and at a lower frequency in other cancers [33]. MSI and TMB are genomic biomarkers used to identify patients who may benefit from immune checkpoint inhibitors [34]. We calculated TMB scores using Perl software (version 5.32.1) and corrected them by dividing by the total length of the exons. Somatic mutation data were captured from the TCGA database, and MSI scores were calculated for all samples. The correlation of MTA2 expression with TMB and MSI was analyzed using Spearman’s rank correlation coefficients [35].

### 2.6. Enrichment Analysis of MTA2 in Pan-Cancer

We downloaded the gene set “c2.cp.kegg.v7.5.1.symbols.gmt” from the Molecular Signatures Database (MSigDB, https://www.gsea-msigdb.org/gsea/msigdb, accessed on 5 July 2022). Gene set enrichment analysis (GSEA) of MTA2 in pan-cancer was performed using the “limma”, “org.Hs.eg.db”, “DOSE”, “clusterProfiler”, and “enrichplot” packages (*p* < 0.05).

### 2.7. Analysis of Genes Co-Expressed with MTA2

We employed Pearson’s correlation analysis [35] to screen 231 genes from 374 TCGA-LIHC patients that were co-expressed with MTA2 (cor > 0.6, *p* < 0.001).

### 2.8. The Association between MTA2 and Immunity in LIHC

According to seven bioinformatics algorithms (TIMER, CIBERSORT, CIBERSORT-ABS, QUANTISEQ, MCPCOUNTER, XCELL, and EPIC) [36], we analyzed the correlation of MTA2 with immune cells in pan-cancer. “tibble”, “survival”, “survminer”, “sva”, “limma”, “DESeq2”, “devtools”, “limSolve”, “GSVA”, “e1071”, “preprocessCore”, “ggplot2”, “biomaRt”, “ggpubr”, “devtools”, “tidyHeatmap”, “caret”, “glmnet”, “ppcor”, “timeROC”, and “pracma” R packages were applied for this analysis. On the SangerBox online website, we applied the Pearson’s correlation coefficient to analyze the correlation of MTA2 with immune checkpoint genes and immune cells in pan-cancer. Using the Cancer Imaging Archive database (TCIA, https://www.tcia.at/home, accessed on 10 October 2022), we analyzed the Immunophenoscore (IPS) [37] of MTA2 in LIHC. Based on the IMvigor210 dataset, we analyzed the immunotherapeutic response of MTA2 to LIHC. Using the “limma” and “estimate” R packages and applying the ESTIMATE algorithm, we analyzed the TME of LIHC and calculated the ESTIMATEScore, StromalScore, and ImmuneScore. Applying Spearman’s correlation method [38], we explored the correlation between MTA2 and immune cells in LIHC (*p* < 0.05).

### 2.9. Single Cell Sequencing Analysis

Four human HCC samples containing 3200 cells from the GEO database (https://www.ncbi.nlm.nih.gov/geo/query/acc.cgi?acc=GSE146115, accessed on 26 August 2022) were included in this study. The “Seurat” and “SingleR” R packages were applied for scRNA-seq data analysis. Subsequently, the “RunTSNE” function was used to analyze the t-distributed stochastic neighbor embedding (t-SNE). Cell clustering was displayed using t-SNE-1 and t-SNE-2.

### 2.10. Screening of Antitumor Drugs

We performed drug screening for LIHC utilizing the Xsum algorithm [39]. The molecular signature of the disease was extracted from the analysis of the expression difference between cancer and para-cancer. Based on this molecular signature, a Connectivity Map (CMap) analysis [40] was performed to find drugs that may potentially fight against LIHC. The lower the score we obtain, the more likely this drug is to reverse the molecular signature of the disease and is theoretically more likely to have the power to treat the disease.

### 2.11. Molecular Docking between MTA2 Protein and MK-886

After screening the antitumor drug MK-886 using the Xsum algorithm, we combined the protein MTA2 (protein ID:O94776) with MK-886 (PubChem CID:3651377) by molecular docking technique [41]. The crystal structure of the MTA2 protein (Protein ID: O94776) was obtained from the PubChem database (https://pubchem.ncbi.nlm.nih.gov/, accessed on 24 October 2022). The 3D structure of MK-886 (PubChem CID: 3651377) was acquired from the PubChem database. The pre-processing of the MTA2 protein and small molecule ligand MK-886 was performed using Discovery Studio 2019 (DS) (BIOVIA Corp, San Diego, CA, USA). The MTA2 protein was dehydrated, and its active site was defined. We then performed a semi-flexible molecular docking process and analyzed the final docked complexes using PyMOL software (version 2.4, https://pymol.org/2/, accessed on 28 October 2022) to generate the 3D binding pattern pictures. The crystal structures of hub protein targets were acquired from the Protein Data Bank (PDB, https://www.rcsb.org/, accessed on 28 October 2022).

### 2.12. Immunohistochemistry

We purchased tissue microarrays containing 89 cases of paired human hepatocellular carcinoma and adjacent normal tissues (HLiveH180Su30) from Shanghai Outdo Biotech Company, China. Briefly, paraffin-embedded sections were deparaffinized in xylene and rehydrated through graded alcohol. Sections were treated with 3% H_2_O_2_ for 10 min. For antigen retrieval, sections were placed in 0.01 M (pH 6.0) citrate buffer and microwaved for 5 min. Rabbit monoclonal antibody to MTA2 (ab171073, Abcam, Boston, MA, USA; 1:200 dilutions) was used as the primary antibody [42]. Subsequent procedures were performed according to the operation instructions of the immunohistochemistry detection kit (PV-6000, ZSGB-BIO, Beijing, China). Positive staining was detected by 3, 3-diaminobenzidine (DAB). The intensity of MTA2 staining was scored as 0 (no signal), 1 (weak), 2 (moderate), and 3 (strong). Immunohistochemistry was performed as previously described, and the percentages of staining were scored as 1, 1–25%; 2, 26–50%; 3, 51–75%; and 4, 76–100%. The scores of each sample were multiplied to give a final score of 0–12, and samples were finally identified: tissues with a staining score greater than five were defined as having high expression, and tissues with a staining score of four or less were defined as having low expression [42].

### 2.13. Cell Culture

HepG2 cells were purchased from Procell Company (Wuhan, China) and cultured with DMEM (Gibco, Carlsbad, CA, USA) medium containing 10% FBS (Gibco, Carlsbad, CA, USA) at 37 °C with 5% CO_2_ and saturated humidity.

### 2.14. CCK-8 Cell Proliferation Assay

HepG2 cells were grown in 96-well plates at an initial density of 5 × 10^3^ cells per well for 24 h. Then cells were treated with different concentrations of MK-886 (MedMol, Shanghai, China) solubilized by DMSO (Solarbio, Beijing, China), with six duplicates for each concentration. The final concentrations of DMSO were less than 1%. After MK-886 treatment for 24 h and 48 h, respectively, CCK-8 reagent (Beyotime, Shanghai, China) was added to each well and incubated in an incubator at 37 °C for 3 h. The optical density (OD) value of each well at 450 nm was detected using a microplate reader (BioTek, Winooski, VT, USA). The cell growth inhibition rate (IR) was then calculated at the following equation: IR = (OD _experimental group_ − OD _blank group_/OD _control group_ − OD _blank group_) × 100%.

### 2.15. Cell Viability Assay

HepG2 cells were treated with different concentrations of MK-886, as described above. After incubation for 48 h, cells were washed once with PBS, and 100 μL of Calcein AM/PI assay working solution (Beyotime, Shanghai, China) was added to each well and incubated for 30 min at 37 °C. The cell viability was observed under a confocal microscope (Olympus, IXplore SpinSR, Tokyo, JPN). Live cells were stained with green fluorescence, while dead cells were stained with red fluorescence.

### 2.16. Statistical Methods

All gene expression data in this study were normalized by log2 transformation. Normal and cancerous tissues were compared using a two-group t test, with *p* < 0.05 indicating statistical significance. Kaplan Meier curves, log-rank tests, and Cox proportional hazards regression models were utilized for all survival analyses in this study. Correlation analysis between two variables was performed applying Spearman’s or Pearson’s test, and *p* < 0.05 was considered significant. All statistical analyses were processed by R software (version 4.2.1, https://www.r-project.org/, accessed on 14 July 2022) and Strawberry Perl software (version 5.32.1, https://strawberryperl.com/). All R packages were installed on the bioconductor (https://www.bioconductor.org/, accessed on 20 July 2022) and git-hub (https://github.com/, accessed on 20 July 2022) websites. We performed the flowcharting for this study utilizing the BioRender website (https://www.biorender.com/, accessed on 2 January 2023).

## 3. Results

### 3.1. Differential Expression of MTA2 between Tumor and Normal Tissues in Pan-Cancer and Mutation of MTA2

The process of this study is shown in Figure 1. We acquired 11,057 pan-cancer and normal samples from 33 tumor types from the TCGA database to investigate the difference in the expression of MTA2 between tumor and normal tissues. The results demonstrated that MTA2 was highly expressed in BLCA, BRCA, CESC, CHOL, COAD, ESCA, GBM, HNSC, KIRC, KIRP, LIHC, LUAD, LUSC, PRAD, READ, SARC, STAD, THCA, and UCEC compared with normal tissues (*p* < 0.05). Our findings indicated that MTA2 was overexpressed in most tumors. (Figure 2A). Meanwhile, our results revealed that MTA2 was generally highly expressed in tumor tissues from the TCGA database (Figure 2B). We speculated that MTA2 might play a crucial role in tumor development. We also analyzed MTA2 expression at different stages of pan-cancer (Figure 2C). Our results suggested that MTA2 was differentially expressed in different tumor stages. Interestingly, the expression of MTA2 was closely related to the staging of ACC, BLCA, LIHC, and MESO (*p* < 0.05). In ACC, MTA2 expression was positively correlated with staging. However, in BLCA and MESO, there was a trend towards an early increase in MTA2 expression. In LIHC, MTA2 expression was positively correlated with stage I-III, but showed a decrease in stage IV. To investigate the regulatory association of MTA2 with other genes in tumors, the PPI networks were obtained from the GeneMANIA (Figure 2D) and comPPI databases (Figure 2E) online websites. The PPI network indicated a strong interaction between MBD3 and MTA2 expression, scoring up to 0.99. A higher score suggested a higher association between the two genes, with a maximum score of 1 (Appendix A). MTA2 and MBD3 might co-regulate tumor development through co-expression and physical interaction in the cell membrane, mitochondria, and nucleus, leading to tumor cell expression and migration. Analysis of MTA2 mutations in pan-cancer in the cBioPortal database revealed that MTA2 was amplified in cholangiocarcinoma, adrenocortical carcinoma, ovarian serous cystadenocarcinoma, liver hepatocellular, mesothelioma, pheochromocytoma, paraganglioma, and lung squamous cell carcinoma (Figure 3A). We believed that MTA2 contributed to the development of these tumors through amplification and generated a poor prognosis, suggesting that MTA2 was capable of triggering tumor progression through amplification. We analyzed the location of MTA2 in human cells and concluded that MTA2 played a regulatory role mainly in the nucleus and cytosol, which can provide hints for the regulatory study of MTA2 in cancer cells (Figure 3B). The cell structure diagram from the GeneCards database showed that MTA2 regulates tumorigenesis and progression predominantly in the nucleus and cytosol. Figure 3C presents the 3D structure of the AlphaFold prediction from the GeneCards database. By utilizing the structure of MTA2, we can perform molecular docking techniques to find the corresponding targeted tumor drugs to inhibit tumor progression and improve patient prognosis. We explored mutations at the MTA2 locus in the cBioProtal database, and the frequency of mutations at these points was shown on the y-axis (Figure 3D). MTA2 contains four distinct regions: a BAH domain from positions 1–144, an ELM2 domain from positions 145–256, a SANT domain from positions 263–315, and an atypical GATA zinc finger domain from positions 367–394. Therefore, MTA2 has a complex structural domain with several potential regulatory regions, and BAH, EML2, SANT, and zinc finger structural domains may be engaged in protein-protein co-regulatory effects. Figure 3E displays the location of MTA2 on the human chromosome in the UCSC database. MTA2 is located on human chromosome 11q12.3.

### 3.2. Prognostic Value of MTA2 in Pan-Cancer

To investigate the association between MTA2 and cancer prognosis, we performed a survival analysis of MTA2 based on the “survminer” and “survival” R packages (*p* < 0.05). According to the median value of MTA2 expression in tumors, we classified the expression of MTA2 into high and low expression groups. Consequently, we concluded that the expression of MTA2 was significantly associated with the OS of ACC (*p* = 0.025), BRCA (*p* = 0.025), ESCA (*p* = 0.04), KIRC (*p* = 0.032), LAML (*p* = 0.007), LIHC (*p* = 0.046), MESO (*p* = 0.016), and STAD (*p* = 0.026) (Figure 4A–H). High MTA2 expression had unfavorable OS in ACC, KIRC, LAML, LIHC, and MESO. However, high expression of MTA2 had favorable OS in BRCA, ESCA, and STAD. This indicated that the effects of MTA2 in diverse tumors were multifaceted, which might be relevant to the tumor microenvironment of different tumors. We concluded that the expression of MTA2 was significantly associated with the DFS, DSS, and PFS of ACC (DFS: *p* < 0.001; DSS: *p* = 0.029; PFS: *p* = 0.004), CHOL (DFS: *p* = 0.016), ESCA (DFS: *p* = 0.021; DSS: *p* = 0.019), OV (DFS: *p* = 0.002; PFS: *p* = 0.004), PRAD (DFS: *p* = 0.043; PFS: *p* = 0.011), KIRC (DSS: *p* = 0.043), STAD (PFS: *p* < 0.001), UCEC (DSS: *p* = 0.01), ESCA (DFS: *p* = 0.021; DSS: *p* = 0.019; PFS: *p* = 0.031), GBM (PFS: *p* = 0.019) and UVM (PFS: *p* = 0.033) (Figure 5). As for DFS, MTA2 with high expression had a shorter DFS in ACC, CHOL, and PRAD. The results indicated that MTA2 overexpression had shorter DSS in ACC and KIRC. Meanwhile, high expression of MTA2 had shorter PFS in ACC, UVM, and PRAD. We also created forest plots to investigate the OS (Figure 6A and Appendix A), DFS (Figure 6B), DSS (Figure 6C), and PFS (Figure 6D) of MTA2 in 33 cancer types. By OS analysis, we found that the prognosis of MTA2 in LIHC was the most significant in 33 cancer types (*p* < 0.001), and the results indicated that high expression of MTA2 in LIHC had a poor prognosis.

### 3.3. Microsatellite Instability, Tumor Mutation Burden and Area under the Curve (AUC) Values of MTA2 in Pan-Cancer

Based on the “fmsb” R package, we applied Spearman’s correlation coefficient to analyze the correlation of MTA2 with MSI in pan-cancer (Figure 7A). The result demonstrated that MTA2 expression correlated significantly with the MSI of UCEC (cor = 0.23, *p* < 0.001), STAD (cor = 0.25, *p* < 0.001), LUAD (cor = 0.16, *p* < 0.001), KICH (cor = 0.36, *p* < 0.01), SARC (cor = 0.17, *p* < 0.01), LUSC (cor = 0.12, *p* < 0.01), COAD (cor = 0.13, *p* < 0.01) and DLBC (cor = −0.36, *p* < 0.05). In particular, we found that UCEC, STAD, SARC, LUSC, LUAD, KICH, DLBC, and COAD were positively correlated with MTA2 expression, while MSI has emerged as an excellent predictor of the sensitivity of immunotherapy-based strategies of COAD and STAD. According to the “fmsb” R package, we also applied Spearman’s correlation coefficient method to analyze the correlation of MTA2 expression in pan-cancer with TMB (Figure 7B). Radar plots revealed a remarkable correlation between MTA2 and TMB of STAD (cor = 0.40, *p* < 0.001), UCEC (cor = 0.33, *p* < 0.001), LUAD (cor = 0.26, *p* < 0.001), THYM (cor = −0.49, *p* < 0.001), COAD (cor = 0.26, *p* < 0.001), BRCA (cor = 0.12, *p* < 0.001), LGG (cor = 0.16, *p* < 0.001), THCA (cor = −0.15, *p* < 0.001), ACC (cor = 0.35, *p* < 0.01), UCS (cor = 0.31, *p* < 0.05), SARC (cor = 0.14, *p* < 0.05) and BLCA (cor = 0.11, *p* < 0.05). By analyzing the association between malignancy and TMB, we can observe the number of cancer mutations. To assess the accuracy of the model for MTA2 in tumors, we plotted radar plots showing the AUC of MTA2 in pan-cancer (Figure 8). Table 1 displayed clearly the AUC values of MTA2 in different tumors. Interestingly, we found that MTA2 had good prognostic value for a variety of tumors. Among them, the AUC value is larger than 0.8 in 14 cancer types: CESC (AUC = 0.838), CHOL (AUC = 1.000), COAD (AUC = 0.859), DLBC (AUC = 0.851), ESAD (AUC = 0.811), GBM (AUC = 0.815), LGG (AUC = 0.871), LIHC (AUC = 0.859), LUSC (AUC = 0.930), PAAD (AUC = 0.973), STAD (AUC = 0.818), TGCT (AUC = 0.804), THYM (AUC = 0.909) and UCEC (AUC = 0.800). It is worth mentioning that the AUC value is equal to 1 in CHOL. The results indicated that MTA2 is a promising indicator for prediction of tumor prognosis.

### 3.4. Analysis of Genes Co-Expressed with MTA2 in LIHC, Pan-Cancer Extent of Immune Infiltration and Enrichment Analysis

Based on the above analysis, we found that MTA2 is overexpressed and is an indicator of poor prognosis with a high AUC value in LIHC from the TCGA database. Moreover, the prognostic value of MTA2 in LIHC was the most significant among 33 cancer types (*p* < 0.05). Therefore, we chose LIHC for further in-depth analysis to investigate the mechanism of MTA2 in HCC. First, on the basis of Pearson’s correlation coefficient analysis (cor > 0.6, *p* < 0.001), we screened 231 genes co-expressed with MTA2 in 374 TCGA-LIHC patients based on the “ggplot2”, “ggpubr”, and “ggExtra” R packages (Figure 9 and Appendix A). These MTA2 co-expressed genes are positively correlated with MTA2 expression in HCC and may interact with MTA2 to contribute to HCC development. According to the “EPIC”, “MCPcounter”, “GSVA”, and “estimate” R packages, we performed an immunoassay for MTA2 in pan-cancer. The analysis of various algorithms demonstrated that the expression of MTA2 in tumors was closely linked to the infiltration of immune cells. Figure 10A illustrates that MTA2 is positively correlated with most immune cells in pan-cancer. Meanwhile, we identified a high relevance of MTA2 to cancer-associated fibroblasts in KIRP, LIHC TGCT, and UVM (EPIC, MCPCOUNTER, TIDE, and XCELL methods). These results also revealed that MTA2 was highly correlated with MDSCs (myeloid-derived suppressor cells) (TIDE method), macrophages/monocytes (MCPCOUNTER, CIBERSORT, CIBERSORT-ABS, QUANTISEQ and XCELL methods) and neutrophils in most cancers. According to the CIBERSORT algorithm, we analyzed the relevance of MTA2 in 22 kinds of immune cells in pan-cancer (Spearman’s correlation method) and demonstrated that MTA2 was positively related to mast cells activated in LIHC (cor = 0.134, *p* < 0.05) (Figure 10B). MTA2 was positively correlated with CD4 T cells in most cancers, as derived through the EPIC algorithm (Figure 10C). CD4 T cells contribute to immunotherapy in cancer, and our findings suggested that MTA2 correlates strongly with CD4 T cells in most tumors, so indirectly modulating the immunotherapeutic role of CD4 T cells in tumors by targeting MTA2 is a promising strategy. We performed a GSEA analysis of MTA2 and its co-expressed genes in pan-cancer based on the “clusterProfiler” R package (Figure 11). The pathway file was from the MSigDB database of the “h.all.v7.5.1.symbols.gmt” file. We observed that these gene sets in BRCA, KIRC, KIRP, and LIHC were significantly enriched in the “Oxidative phosphorylation” (OXPHOS) pathway. BRCA, KIRC, KIRP, and LIHC are significantly enriched in the OXPHOS pathway, suggesting that OXPHOS inhibitors have the potentiality to treat these tumors or alleviate tumor hypoxia to improve treatment outcomes. The “Fatty acid metabolism” pathway was remarkably enriched in KIRC, KIRP, and LIHC. Therefore, restricting the y of fatty acids to limit the proliferation of cancer cells is probably a suitable treatment option for these tumors. Meanwhile, we noted that the “PI3K-AKT-mTOR” and “IFN-alpha response” signaling pathways were significantly enriched in LIHC, suggesting that MTA2 may participate in the regulation of the tumor immune microenvironment in LIHC. The results of the analysis allow for speculation that targeting the PI3K-AKT-mTOR signaling pathway through cancer prevention and treatment mechanisms could improve the treatment of HCC. The LIHC was also enriched for the “Complement”, “Coagulation”, “Cholesterol homeostasis”, “Bile acid metabolism”, and “Adipogenesis” pathways. Subsequently, we applied the “org.Hs.eg.db”, “DOSE”, “clusterProfiler”, and “enrichplot” R packages to the enrichment analysis of the Kyoto Encyclopedia of Genes and Genomes (KEGG) pathway of MTA2 in pan-cancer based on the “c2.cp.kegg.v7.5.1.symbols.gmt” file. (*p* < 0.05, FDR < 0.25). The results indicated that MTA2 was enriched in immune-related pathways in many cancers (Figure 12). The “KEGG antigen processing and presentation pathway” was significantly enriched in CHOL (NES = 1.58, *p* < 0.05), the “KEGG pathways in cancer” (NES = 1.32, *p* < 0.05) and the “KEGG T cell receptor signaling pathway” (NES = 1.29, *p* < 0.05) in KIRC, the “KEGG T cell receptor signaling pathway” (NES = 1.37, *p* < 0.05) in THCA, and the “KEGG T cell receptor signaling pathway” (NES = 1.49, *p* < 0.05) and “KEGG regulation of autophagy” (NES = 1.43, *p* < 0.05) in UVM (Appendix A). These results suggested that MTA2 participated in the immune regulation of cancers, which was consistent with our previous GSEA.

### 3.5. Analysis of High-MTA2 and Low-MTA2 Groups in LIHC with Immune Checkpoints, TME Score, IPS and Correlation with Immune Cells

We first performed a differential analysis of MTA2 expression between LIHC and adjacent nontumorous tissues based on the median value (Figure 13A). The results demonstrated that MTA2 was overexpressed in LIHC (*p* < 0.001), and high expression of MTA2 may contribute to the development of LIHC and result in a poor prognosis. Subsequently, we further investigated the associations between MTA2 and immune genes and analyzed the differences in immune gene expression between high-MTA2 and low-MTA2 groups. We found that MTA2 was positively correlated with T cell follicular helper infiltration at LIHC (Figure 13B) (r = 0.3, *p* < 0.05). As the T cell follicular helper cells are CD4 T cells specifically designed to assist B cells, this result echoes the previous results of MTA2 with CD4 T cells in LIHC, indicating that MTA2 expression is closely correlated with CD4 T cells. We analyzed the correlation between MTA2 and alpha fetoprotein (AFP) in LIHC and found that AFP+ LIHC patients had higher MTA2 expression (Figure 13C). The results revealed that high expression of MTA2 in HCC was tightly associated with AFP levels in HCC. We analyzed the correlation of MTA2 with human leukocyte antigen (HLA) genes in LIHC, and the results indicated that MTA2 was positively correlated with HLA gene expression (Figure 13D). The results showed that MTA2 was associated with the immune function of HCC. IMvigor210 recorded expression data from patients who responded or did not respond to anti-PD-L1 immunotherapy. Through the IMvigor210 dataset, we analyzed the immunotherapeutic response of MTA2 in LIHC. The results suggested that the expression of MTA2 was higher in the response group (*p* = 0.02) (Figure 13E). Therefore, we speculated that the high expression of MTA2 in LIHC may be a target for immunotherapy. Subsequently, we analyzed the correlation of MTA2 with immune checkpoint genes in LIHC. The results illustrated that MTA2 was positively associated with immune checkpoint genes (Figure 13F). Further results also illustrated that CTLA-4, HAVCR2, PD-1, and PD-L1 (Figure 13G–J) were highly expressed in the high-MTA2 group, except PD-L2 (Figure 13K) (*p* < 0.001). These results suggested that MTA2 may be a potential target for HCC immunotherapy due to the analyses of MTA2 with immune checkpoint genes and anti-PD-L1 immunotherapy in the IMvigor210 dataset. Furthermore, we constructed an IPS for MTA2 in LIHC and observed that the high MTA2 group scored lower in the CTLA4-negative/PD-1-negative group (*p* = 0.016) (Appendix A). The IPS data of HCC were from the TCIA database.

Subsequently, we also analyzed the relevance of MTA2 and immune genes in pan-cancer (Figure 14A). The results suggested a strong interaction of MTA2 with stimulatory and inhibitory immune genes in most cancers. Remarkably, among the inhibitory immune genes, CD274, HAVCR2, PDCD1, and CTLA-4 were significantly associated with most tumors, especially in LIHC, which is consistent with our previous results (*p* < 0.05). Also among the stimulatory immune genes, CD28, CD80, IL1A, IL1B, and CD40 are strongly associated with most tumors (*p* < 0.05). The above results indicated that the regulation of MTA2 plays a pivotal role in immune function in pan-cancer. We analyzed the correlation of MTA2 with relevant immune cells (B cells, CD4 T cells, CD8 T cells, Neutrophils, Macrophages, and DCs) in pan-cancer and demonstrated that TCGA-LIHC (B cells: cor = 0.38, *p* < 0.0001; CD4 T cells: cor = 0.38, *p* < 0.0001; CD8 T cells: cor = 0.25, *p* < 0.0001; Neutrophils: cor = 0.54, *p* < 0.0001; Macrophages: cor = 0.45, *p* < 0.0001; DCs: cor = 0.51, *p* < 0.0001) had the strongest correlation with immune cells. And we could observe a positive correlation between T cell CD4 and 22 cancer types, with the highest correlation being with KIRC (cor = 0.58, *p* < 0.0001) (Figure 14B). This indicated that MTA2 has a substantial association with T cells CD4 in pan-cancer and that MTA2 expression plays an indispensable role in the immune function of pan-cancer.

Besides, MTA2 expression was positively correlated with ESTIMATEScore (r = 0.16, *p* < 0.01), StromalScore (r = 0.13, *p* = 0.01), and ImmuneScore (r = 0.16, *p* < 0.01) (Figure 15A–C) in TCGA-LIHC. Therefore, we believed that MTA2 plays a prominent role in the tumor immune microenvironment in HCC. Figure 15D revealed that the high-MTA2 group had higher TME scores in the TCGA-LIHC ImmuneScore (*p* < 0.01) and ESTIMATEScore (*p* < 0.05). We calculated the difference in the immune cell infiltration fraction between the high and low MTA2 groups in LIHC and discovered that the majority of the immune cell fraction was higher in the high MTA2 group (Figure 15E). We investigated the correlation of MTA2 with immune cells in LIHC (Figure 15F) and concluded that MTA2 expression was positively correlated with follicular helper T cells (r = 0.3, *p* = 3.2 × 10^−7^), memory activated CD4 T cells (r = 0.18, *p* = 0.0023), memory B cells (r = 0.18, *p* = 0.0035), resting dendritic cells (r = 0.16, *p* = 0.0094), M0 macrophages (r = 0.15, *p* = 0.0098), plasma cells (r = 0.13, *p* = 0.029), and neutrophils (r = 0.12, *p* = 0.04) and negatively correlated with activated NK cells (r = −0.17, *p* = 0.0039), gamma delta T cells (r = −0.2, *p* = 0.001) and M2 macrophages (r = −0.3, *p* = 5.6 × 10^−7^) (Figure 15G). Among them, MTA2 has the highest correlation with T cells follicular helpers in HCC. MTA2 is greatly associated with immune cells and is a vital player in regulating the immune microenvironment in HCC. According to the “ggcor”, “ggplot2”, and “GSVA” R packages, we analyzed the correlation of MTA2 with immune cells and immune function in LIHC (Figure 16A). As shown in the figure, the left side represents immune function, and the right side represents immune cells. The results suggested a strong correlation between the MTA2 and immune function in LIHC.

### 3.6. Single Cell Sequencing and Drug Therapy in HCC

To investigate HCC at the single cell level, we acquired a comprehensive transcriptional profile of 3200 single cells from 4 HCC samples (GSE146115) in the GEO database. We analyzed the distribution type and density of immune cells in HCC using t-SNT plots (Figure 16B,C). Combined with the cell distribution and density distribution maps, the results displayed that the expression level of T cells was higher in these HCC samples, indicating that the expression of MTA2 in HCC was significantly correlated with T cells. Further analysis demonstrated that MTA2 expression was significantly upregulated in hepatocytes and was also enriched in T cells (Figure 16D). To find effective drugs against HCC, we used the Xsum algorithm for drug analysis. We performed differential analysis of HCC and adjacent non-tumor tissues using the R package “limma” (|log2 FC| >1, *p* < 0.05). We then identified 1042 genes with differences and used them to screen for antitumor drugs by molecular characterization in HCC and adjacent non-tumorous tissues. Our CMap analysis showed that MK-886 scored the lowest. Based on the theory of CMap, MK-886 was the most likely drug to treat HCC (Figure 16E). Therefore, we believed that MK-886 has the potential to improve the prognosis of HCC patients as a clinical drug for the treatment of HCC. The R packages “PharmacoGx”, “CoreGx”, “shinyjs”, “shinydashboard”, “magicaxis”, “lsa” and “relation” were used for the Xsum analysis.

### 3.7. Molecular Docking

On the basis of the antitumor drug MK-886 screened by the Xsum algorithm, we evaluated the binding capacity between the MTA2 protein and the small molecule ligand MK-886 via a molecular docking technique. The 3D combination mode of the interaction between MTA2 and MK-886 is shown in Figure 16F. The results demonstrated that the MTA2 protein binds to the MK-886 protein mainly through Π-alkyl interactions (hydrophobic interactions) and alkyl interaction forces. For the small molecule ligand MK-886 and the receptor protein MTA2, the binding energy after docking is −17.0449 kcal/mol. The binding energy is less than or equal to −5 kcal/mol, indicating binding activity. The MTA2 protein is semiflexible bound through the side chain and terminal ligand (MK-886), and the residue PRO-635 of the left MTA2 forms a Π-alkyl interaction (hydrophobic interaction) and alkyl interaction force with the small molecule ligand of protein MK-886. Similarly, residues ALA-628, MET-625, ARG-617, LEU-620, PR0-635, and LEU-616 of MTA2 all form Π-alkyl interactions and an alkyl interaction forces with the small molecule ligand MK-886. Finally, the interaction forces after docking in this complex model revealed that the two molecules mainly rely on alkyl interaction forces to maintain binding. The two-dimensional plane structure model of the interaction between MTA2 protein residues and the drug small molecule MK-886 is shown in Figure 16G.

### 3.8. Construction of the Nomogram and Validation on the Basis of Clinical Information in LIHC

According to seven bioinformatics algorithms (TIMER, CIBERSORT, CIBERSORT-ABS, QUANTISEQ, MCPCOUNTER, XCELL, and EPIC), we analyzed the relationship between MTA2 expression and clinical factors (gender, grade, age, ImmuneScore, ESTIMATEScore, N stage (Lymph Node), M stage (Metastasis), T stage (Tumor), and tumor AJCC stage) in immune-related cells in LIHC (Figure 17A). The results showed a strong correlation between the expression of MTA2 and immune cells in HCC. We established a nomogram based on MTA2 expression and this clinical information in LIHC (Figure 17B). The 1-, 3- and 5-year calibration curves revealed the wonderful predictive power of the nomogram (Figure 17C). The results concluded that the nomogram constructed based on the expression of MTA2 and clinical characteristics of HCC can effectively predict the prognosis of HCC patients, and MTA2 can be regarded as a useful indicator to determine the prognosis of HCC patients for the clinical therapy of HCC.

### 3.9. Analysis of Mutations in High and Low MTA2 Expression Groups in HCC

By analyzing the cBioPortal database, we found that the mutation frequency of MTA2 in pan-cancer was 1.5% (*p* < 0.05) (Appendix A). Furthermore, we compared the mutations of the high-MTA2 group and the low-MTA2 group in LIHC, which resulted in the 20 genes (TP53, LRP1B, OBSCN, RB1, NBEA, SPEG, DNAH10, TSC2, FAM47A, SI, SYCP2, CTNNA2, ATP8A2, IL6ST, TENM2, COL7A1, DNAH14, CLTC, NLRP12, and KPRP) with the highest mutation frequencies, and concluded that the two groups had the highest frequency of missense mutations in the SangerBox online website (Appendix A). Among them, the frequency of TP53 mutations was as high as 46.8%, indicating that TP53 mutations were usually intimately associated with a poorer prognosis in LIHC. We analyzed the point mutations of MTA2 in pan-cancer from the SangerBox online website (Appendix A). The results revealed that COAD (4.3%) had the highest mutation rate.

### 3.10. MTA2 Protein Is Overexpressed and Associated with Poor Prognosis in Human HCC

We examined the protein expression of MTA2 in 89 pairs of HCC tissues and normal tissues by immunohistochemistry. The representative results showed high MTA2 expression in the HCC tissue and low MTA2 expression in the paired adjacent normal tissue (Figure 18A,B). Further analysis revealed that the MTA2 protein was up-regulated in HCC tissues (Figure 18C) (*p* < 0.001). Survival analysis demonstrated that patients with high expression of MTA2 in HCC had poorer survival (Figure 18D) (*p* < 0.001). Additionally, we compared the survival of the four T stages and indicated that among HCC patients T_4_ had the worst survival and T_1_ had the best survival (Figure 18E) (*p* < 0.001). Table 2 displays the association of clinicopathological characteristics with MTA2 expression in 89 patients with HCC (*p* > 0.05).

### 3.11. Growth Inhibition Effects of MK-886 on Proliferation of HCC Cells

We utilized the CCK-8 assay to detect the effect of MK-886 on the proliferation of cell. As shown in Figure 18F, MK-886 treatment inhibited the proliferation of HepG2 cells in a concentration- and time-dependent manner. The IC_50_ values were 42.09 μM at 24 h and 20.41 μM at 48 h. These results indicated that low doses of MK-886 were able to inhibit the proliferation of HepG2 cells. As shown in Figure 18G, MK-886 could affect the viability of HepG2 cells. As the drug concentration increased, the proportion of living cells decreased and dead cells increased gradually. Taken together, the results suggested that MK-886 had a remarkable inhibitory effect on the growth of HepG2 cells.

## 4. Discussion

Cancer poses a serious threat to human health due to its high morbidity [23] and mortality [43]. Common cancer treatments include surgical resection, radiotherapy, and chemotherapy, but the efficacy of some malignancies remains poor [44]. Early detection and effective treatment can improve the prognosis of cancer patients [45]. Pan-cancer analysis is capable of revealing the similarities and differences between various cancers, providing new insights into the design of cancer prevention and personalized treatment strategies. MTA2 has been reported to be frequently genetically amplified in human cancers, and upregulation of MTA2 could promote cancer metastasis and progression [17]. Our study demonstrated that MTA2 was highly expressed in 19 out of 33 types of cancer. The results for BRCA, COAD, ESCA, GBM, KIRC, LIHC, LUAD, OV, PAAD, PRAD, STAD, and THCA were similar to previous studies [21,22,23,24,25,46,47,48,49,50,51,52,53,54,55]. We found that MTA2 overexpression was correlated with both good and poor tumor prognosis, depending on cancer type, indicating the multifaceted roles of MTA2 in different tumors, which might be related to the diverse tumor microenvironment of various tumors. The MTA2 expression level is closely related to the prognosis of tumors, and it has been reported that higher MTA2 expression is related to an adverse prognosis in pancreatic cancer patients [21]. Kaplan–Meier survival analysis revealed that high MTA2 expression was associated with poor prognosis in ACC, KIRC, LAML, LIHC, and MESO. In contrast, high MTA2 expression was associated with a good prognosis in patients with BRCA, ESCA, and STAD. For validation of the prognostic value of MTA2 in LIHC, we performed immunohistochemical experiments. We found that MTA2 is up-regulated in HCC compared with normal tissues, and high MTA2 expression is associated with poor survival in HCC patients. Our immunohistochemistry results further confer MTA2 as a reliable adverse prognostic marker for HCC. In contrast, high MTA2 expression was associated with a good prognosis in patients with BRCA, ESCA and STAD.

Interestingly, our enrichment analysis revealed that the interferon alpha response pathway was significantly enriched in HCC. More importantly, we discovered that MTA2 overexpression was associated with cancer immunity, TMB and MSI. We performed bioinformatics analysis to identify the GSEA pathway associated with MTA2 expression. We found that MTA2 was positively associated with Type 2 helper T cells (a subset of CD4+ T cells) in most cancers. In recent years, immunotherapy has exhibited greater efficacy in dealing with cancer. The results of this study concluded that the expression of MTA2 was associated with the degree of immune infiltration in a variety of cancers. HCC was used as an example for illustration. We discovered that MTA2 expression levels were significantly correlated with T follicular helper cells, activated memory CD4 T cells, memory B cells, resting dendritic cells, M0 macrophages, plasma cells, neutrophils, activated NK cells, gamma delta T cells, and M2 macrophages, suggesting a close link between MTA2 expression and tumor immunity. TMB is a promising biomarker for pan-cancer prediction [56], which can lead immunotherapy into the era of precision medicine [57]. It has been shown that TMB is positively correlated with the outcome of ICIs treatment for most cancers [6]. TMB has been proven to be a useful biomarker for immune checkpoint blockade (ICB) selection in certain cancer types [58]. MSI is an essential biomarker in ICIs treatment. MSI-H in colorectal cancer is an independent predictor of clinical features and prognosis [59]. Our study indicated that MTA2 expression was positively correlated with TMB in 12 cancer types and with MSI in 8 cancer types. This may suggest that MTA2 expression levels affect TMB and MSI in cancers, thereby influencing the response of cancer patients to immune checkpoint inhibition therapy. Ultimately, this will provide new insights into the prognosis of immunotherapy. From existing studies and our findings, we speculated that cancer patients with high MTA2 expression and high TMB and MSI would probably have a better prognosis after ICIs treatment.

The interaction between tumor cells and the tumor microenvironment plays a decisive role in tumor progression, metastasis, and response to therapy. The tumor microenvironment has attracted great research and clinical interest as a therapeutic target for cancer [60]. Our results revealed that MTA2 was closely related to the TME of LIHC. We discovered that LIHC with high MTA2 expression had higher TME scores, and MTA2 expression was positively correlated with TME scores in LIHC. It was reported that MTA2 can interact with HDAC2/CHD4 to affect the TME and thus promote LIHC formation and progression [61]. Meanwhile, loss of MTA2 function may impair B-cell development and lead to immune system defects [62]. Furthermore, inactivation of MTA2 leads to abnormal T-cell activation and lupus-like autoimmune disease in mice [63]. In this study, we found that MTA2 was significantly enriched in antigen processing and presentation, cancer, T-cell receptor signaling, and the regulation of autophagy pathways in cancer via enrichment analysis. This meant that MTA2 probably influences cancer progression by regulating immune pathways. In summary, combining previous reports and our results, we speculated that MTA2 could be a target for cancer immunotherapy.

The development of anticancer therapies prompts scientists to discover new drugs [64]. With the development of scRNA-seq and increasing research on human cancers, the cellular heterogeneity, immune landscape, and pathogenesis of different cancers have been uncovered step by step [65]. Upregulation of MTA2 expression in hepatocytes and T cells, as demonstrated by scRNA-seq analysis. Through bioinformatics (Xsum algorithm) [39], precise mode of action, and computational methods, we discovered the drug MK-886 (5-lipoxygenase inhibitor) that targets HCC. We constructed the CMap score from the molecular signature of the disease. The algorithm determined the optimal parameters to utilize when performing the CMap analysis. It is well documented that MK-886 induces apoptosis and can be used as an antitumor agent in gastric, pancreatic, breast, and lung cancers [66,67,68,69,70,71,72,73,74,75]. Importantly, the lipid-metabolizing enzyme 5-LOX and its metabolite LTB4 have been reported to activate the transcription factor NF-κB in hepatocellular carcinoma cells. MK-886 can inhibit the activation of the NF-κB pathway in hepatocellular carcinoma HepG2 cells through downregulation of p65 protein expression [76]. In addition, MK-886 was identified to be effective in preventing liver injury both through reducing apoptosis and oxidative stress in a hepatic ischaemia–reperfusion injury model [77,78]. To investigate whether MK-886 can bind with the MTA2 protein to target HCC treatment, we performed molecular docking analysis and found that the two molecules have a great docking effect. Moreover, we found that MK-886 significantly inhibited the proliferation of hepatocellular carcinoma HepG2 cells and induced cell death at a low dose. The findings indicated that MK-886 could be a potential drug in HCC treatment and is worthy of being further explored. In this study, we combined multiple bioinformatics approaches, including transcriptome analysis, single cell-level sequencing, and molecular docking techniques. To the best of our knowledge, this is an original study focusing on the roles of MTA2 in multiple cancers (33 types).

## 5. Conclusions

This study provides a prospective view on the importance of MTA2 in cancer immunotherapy, revealing correlations between basic immune indicators and MTA2, which may facilitate the understanding of the immune system and the underlying mechanisms of MTA2. We also applied forefront algorithms for antitumor drug screening in HCC, providing new insights for the clinical treatment of HCC.

## Figures and Tables

**Figure 1 biomolecules-13-00883-f001:**
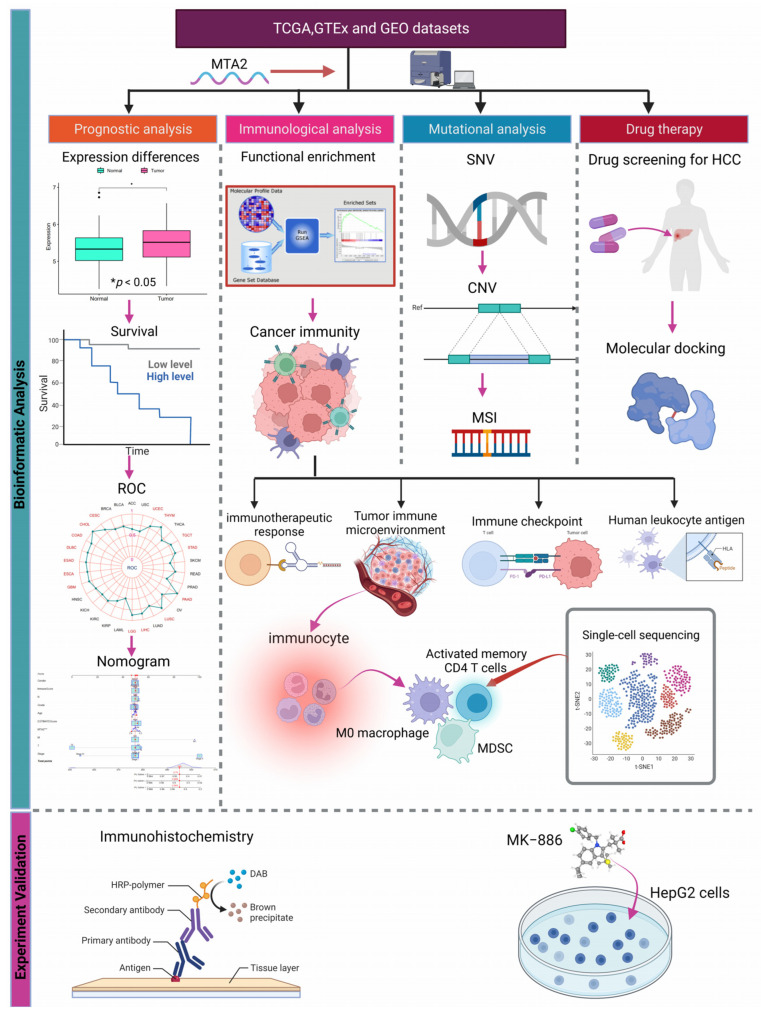
The flowchart of this study.

**Figure 2 biomolecules-13-00883-f002:**
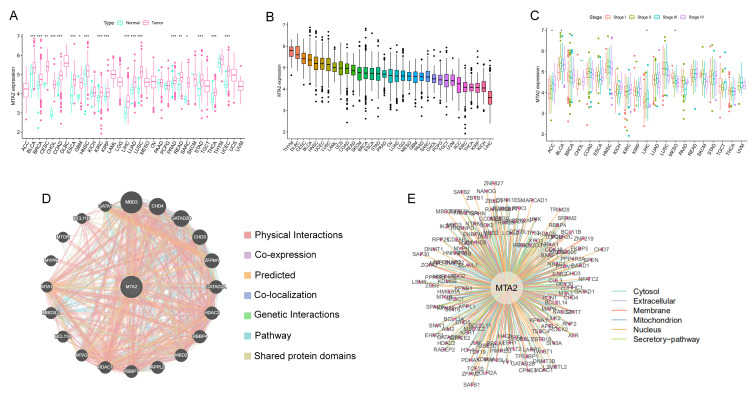
Expression of MTA2 in pan−cancer and construction of PPI networks. (**A**) Comparison of MTA2 expression between tumor and normal samples in pan−cancer. (**B**) MTA2 expression in pan-cancer. (**C**) MTA2 expression in differential tumor AJCC stage. (**D**,**E**) Determination of genes interacting with MTA2 in tumors by PPI network from GeneMANIA and comPPI databases. * *p* < 0.05, ** *p* < 0.01, *** *p* < 0.001.

**Figure 3 biomolecules-13-00883-f003:**
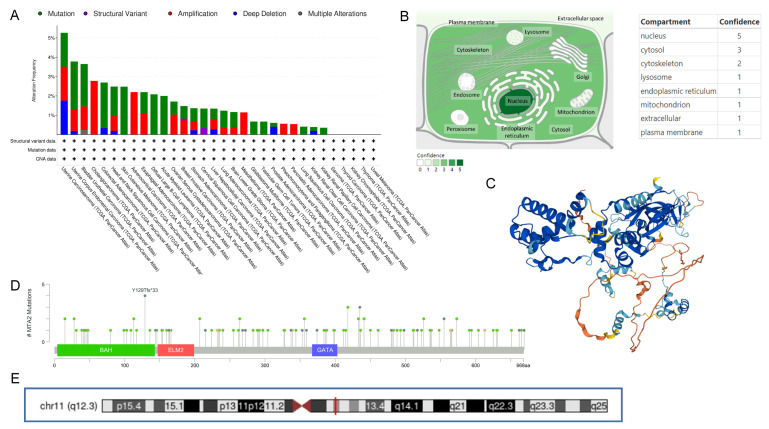
Mutational characteristics of MTA2 in different tumors and its subcellular location. (**A**) Copy number variation of MTA2 in different tumors. Green represents mutation, purple represents structural variant, red represents amplification, blue represents deep deletion, gray represents multiple alteration (cBioPortal database). (**B**) Subcellular location profile of MTA2 in human cells (GeneCards database). (**C**) Predicted three−dimensional structure of MTA2 from AlphaFold (GeneCards database). (**D**) Mutations of MTA2 in the pan−cancer profile; most of them are missense mutations (cBioPortal database). (**E**) Location of MTA2 on human chromosomes (UCSC database).

**Figure 4 biomolecules-13-00883-f004:**
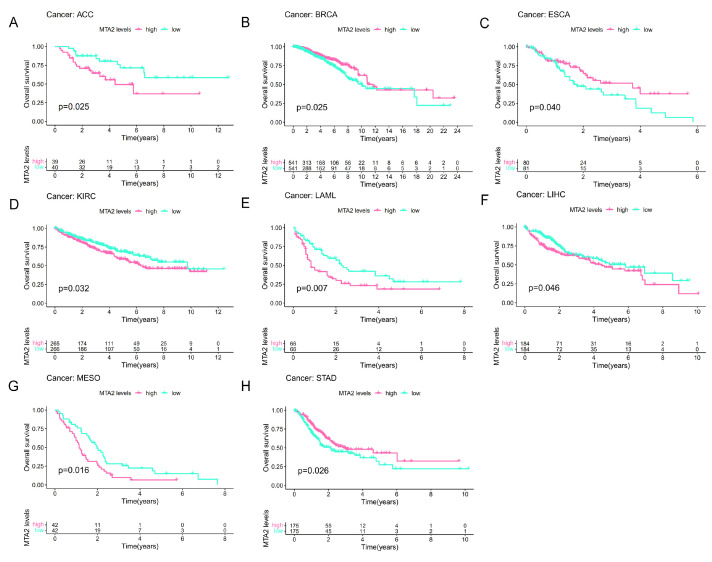
Association between MTA2 expression and overall survival (OS). (**A**–**H**) Kaplan−Meier analysis of the association between MTA2 expression and OS (ACC: *p* = 0.025, BRCA: *p* = 0.025, ESCA: *p* = 0.04, KIRC: *p* = 0.032, LAML: *p* = 0.007, LIHC: *p* = 0.046, MESO: *p* = 0.016, STAD: *p* = 0.026).

**Figure 5 biomolecules-13-00883-f005:**
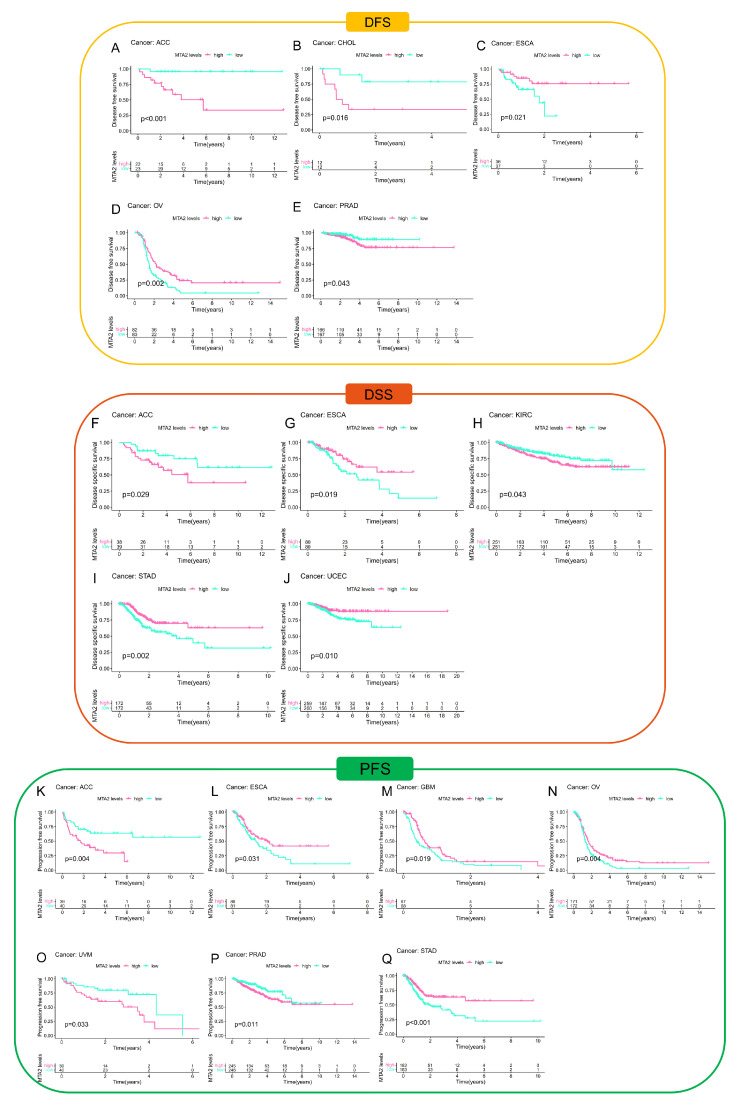
MTA2 expression associate with DFS, DSS, and PFS in pan−cancer. (**A**–**E**) Kaplan−Meier analysis of the association between MTA2 expression and DFS (ACC: *p* < 0.001; CHOL: *p* = 0.016; ESCA: *p* = 0.021; OV: *p* = 0.002; PRAD: *p* = 0.043). (**F**–**J**) Kaplan−Meier analysis of the association between MTA2 expression and DSS (ACC: *p* = 0.029; ESCA: *p* = 0.019; KIRC: *p* = 0.043; STAD: *p* = 0.002; UCEC: *p* = 0.01). (**K**–**Q**) Kaplan−Meier analysis of the association between MTA2 expression and PFS (ACC: *p* = 0.004; ESCA: *p* = 0.031; GBM: *p* = 0.019; OV: *p* = 0.004; PRAD: *p* = 0.011; STAD: *p* < 0.001; UVM: *p* = 0.033).

**Figure 6 biomolecules-13-00883-f006:**
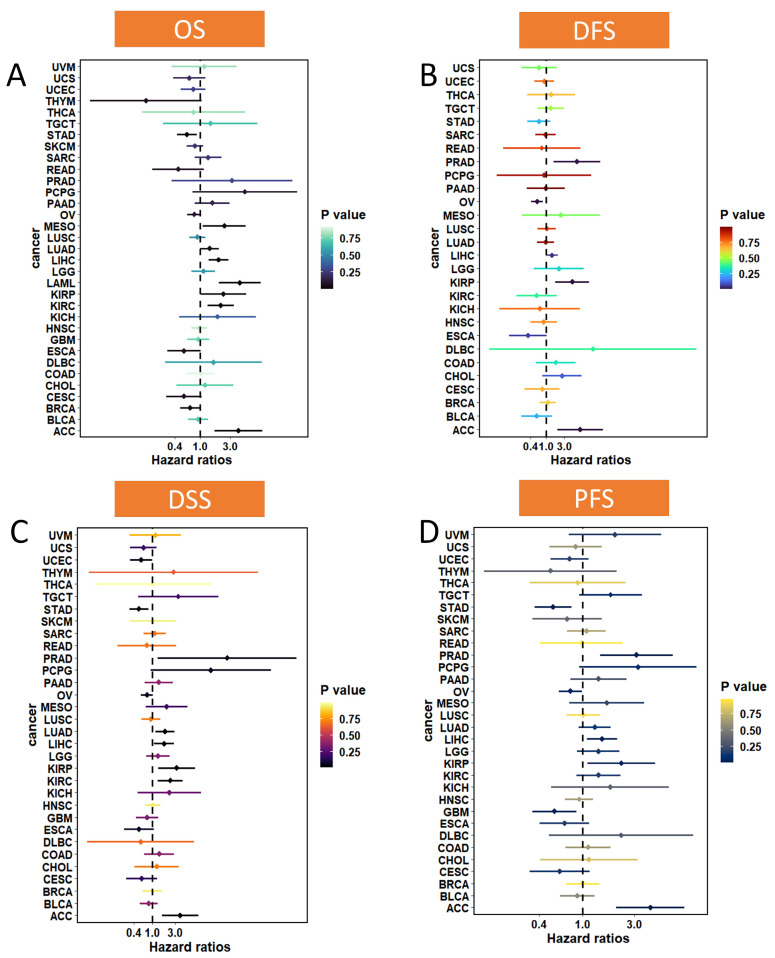
Survival profile of MTA2 in pan−cancer. (**A**) Overall survival (OS). (**B**) Disease−free survival (DFS). (**C**) Disease−special survival (DSS). (**D**) Progression−free survival (PFS).

**Figure 7 biomolecules-13-00883-f007:**
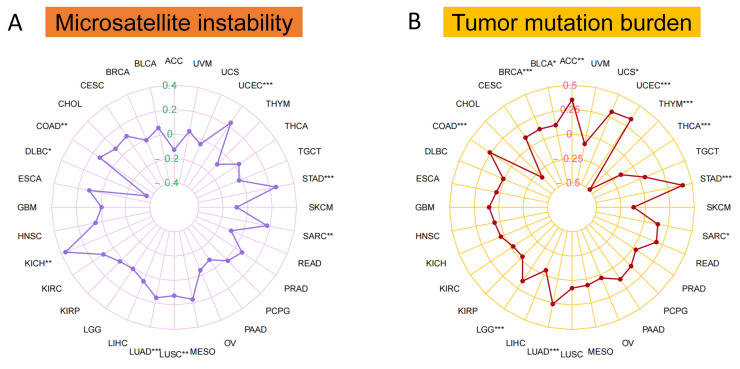
(**A**) Microsatellite instability (MSI). (**B**) Tumor mutation burden (TMB). * *p* < 0.05, ** *p* < 0.01, *** *p* < 0.001.

**Figure 8 biomolecules-13-00883-f008:**
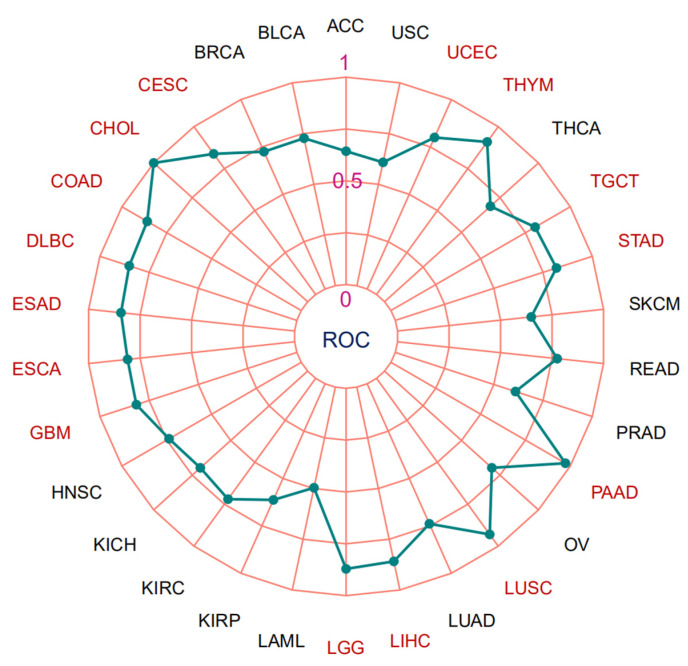
The area under the curve (AUC) values of MTA2 in pan-cancer. (ACC: AUC = 0.643; BLCA: AUC = 0.727; BRCA: AUC = 0.725; CESC: AUC = 0.838; CHOL: AUC = 1.000; COAD: AUC = 0.859; DLBC: AUC = 0.851; ESAD: AUC = 0.811; GBM: AUC = 0.815; HNSC: AUC = 0.737; KICH: AUC = 0.697; KIRC: AUC = 0.720; KIRP: AUC = 0.613; LAML: AUC = 0.496; LGG: AUC = 0.871; LIHC: AUC = 0.859; LUAD: AUC = 0.739; LUSC: AUC = 0.930; OV: AUC = 0.697; PAAD: AUC = 0.973; PRAD: AUC = 0.610; READ: AUC = 0.775; SKCM: AUC = 0.649; STAD: AUC = 0.818; TGCT: AUC = 0.804; THCA: AUC = 0.688; THYM: AUC = 0.909; UCEC: AUC = 0.800; USC: AUC = 0.608). Red section represents tumors with AUC > 8.

**Figure 9 biomolecules-13-00883-f009:**
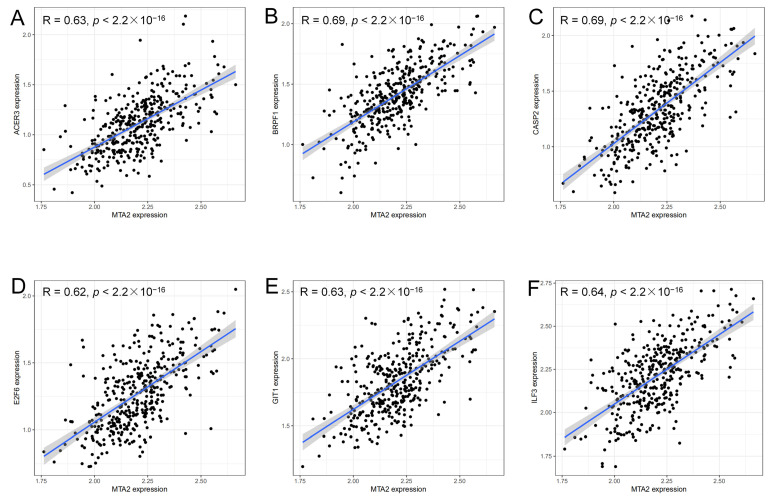
Genes positively associated with MTA2 expression in HCC (cor > 0.6, *p* < 0.001). ACER3 (R = 0.63, *p* < 2.2 × 10^−16^) (**A**), BRPF1 (R = 0.69, *p* < 2.2 × 10^−16^) (**B**), CASP2 (R = 0.69, *p* < 2.2 × 10^−16^) (**C**), E2F6 (R = 0.62, *p* < 2.2 × 10^−16^) (**D**), GIT1 (R = 0.63, *p* < 2.2 × 10^−16^) (**E**) and ILF3 (R = 0.64, *p* < 2.2 × 10^−16^) (**F**) are positively correlated with MTA2 expression in TCGA−LIHC.

**Figure 10 biomolecules-13-00883-f010:**
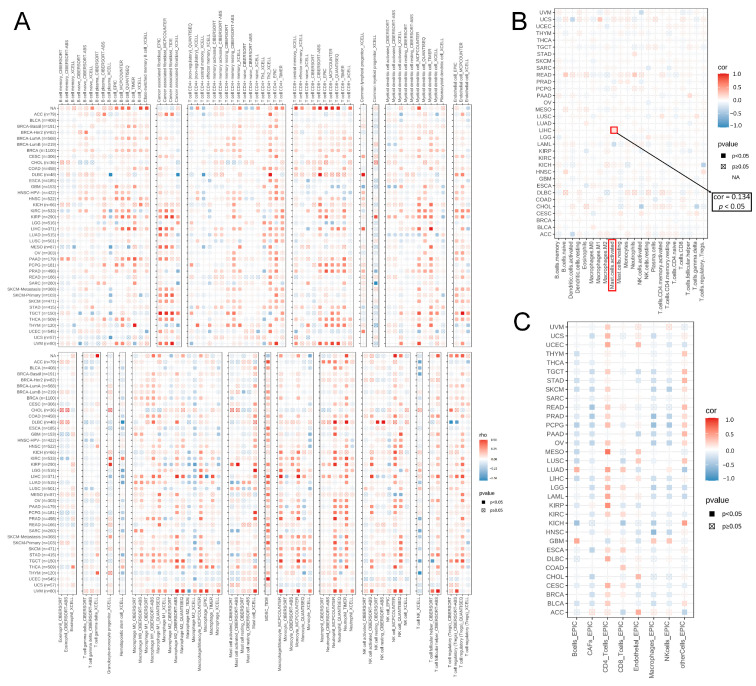
Application of different immune−related bioinformatics methods to analyze the relationship between MTA2 and immune cells in pan−cancer. (**A**) TIMER analysis. (**B**) CIBERSORT analysis. MTA2 was positively related to mast cells activated in LIHC (cor = 0.134, *p* < 0.05). (**C**) EPIC analysis. Red represents positive regulation, and blue represents negative regulation.

**Figure 11 biomolecules-13-00883-f011:**
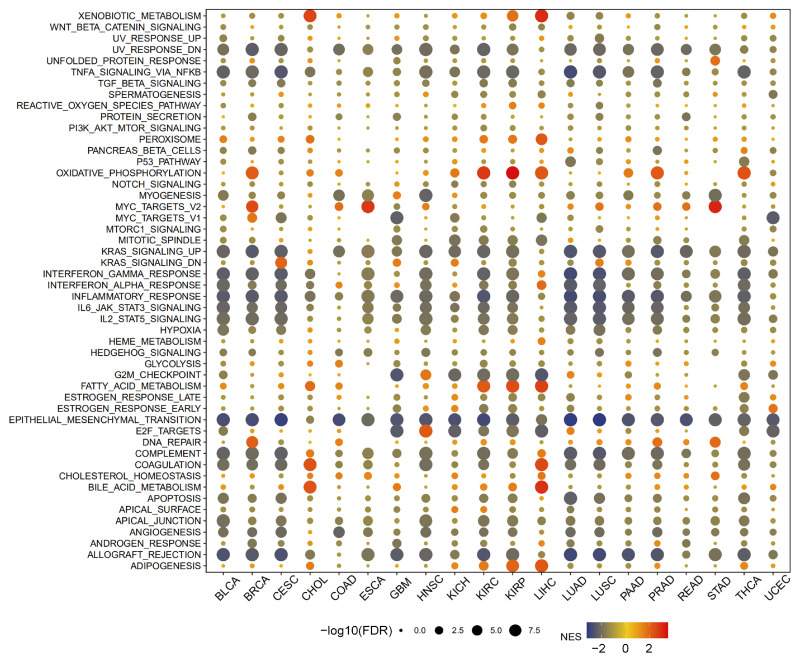
Gene set enrichment analysis (GSEA) of MTA2 in pan−cancer. GSEA revealed significant enrichment of “Oxidation phosphorylation” (BRCA, KIRC, KIRP, LIHC, PRAD and THCA) and “Fatty acid metabolism” pathways (CHOL, KIRC KIRP and LIHC) in multiple tumors.

**Figure 12 biomolecules-13-00883-f012:**
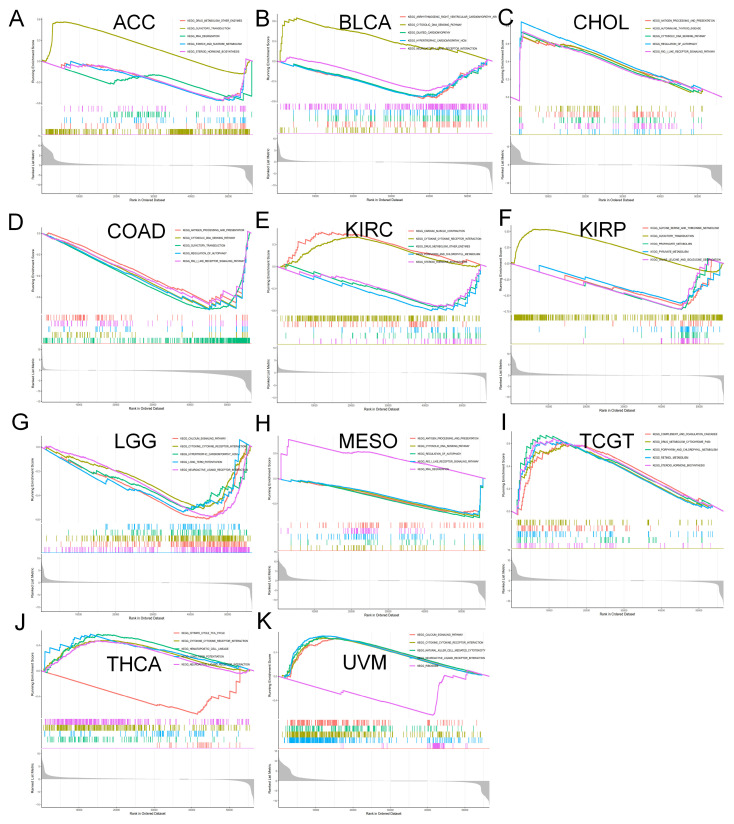
KEGG enrichment analysis of pan−cancer. (**A**) ACC. (**B**) BLCA. (**C**) CHOL. (**D**) COAD. (**E**) KIRC. (**F**) KIRP. (**G**) LGG. (**H**) MESO. (**I**) TCGT. (**J**) THCA. (**K**) UVM.

**Figure 13 biomolecules-13-00883-f013:**
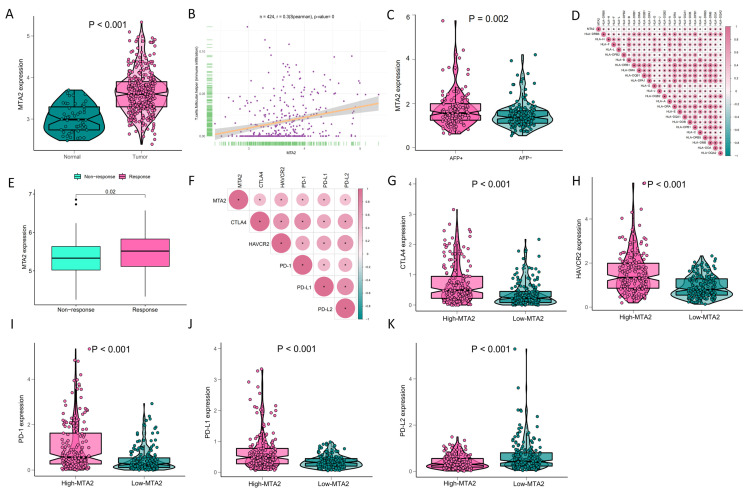
Response of the high−MTA2 and low−MTA2 groups to HCC immunotherapy. (**A**) Differential expression of MTA2 between HCC and adjacent nontumorous tissues (*p* < 0.001). (**B**) Correlation of MTA2 in HCC with follicular helper T cells (immune infiltration) (r = 0.3, *p* < 0.05). (**C**) Relationship between alpha fetoprotein (AFP) and MTA2 expression (*p* = 0.002). (**D**) Analysis of the correlation between MTA2 and HLA genes in HCC. The results revealed a strong association between MTA2 and HLA genes. (**E**) Application of IMvigor210 immunotherapy resulted in a significant difference in MTA2 expression between the responsive and nonresponsive groups in HCC (*p* = 0.02). (**F**) Relevance of MTA2 and immune checkpoints genes in HCC, (**G**) CTLA−4 (*p* < 0.001), (**H**) HAVCR2 (*p* < 0.001), (**I**) PD−1 (*p* < 0.001), (**J**) PD−L1 (*p* < 0.001), (**K**) PD−L2 (*p* < 0.001).

**Figure 14 biomolecules-13-00883-f014:**
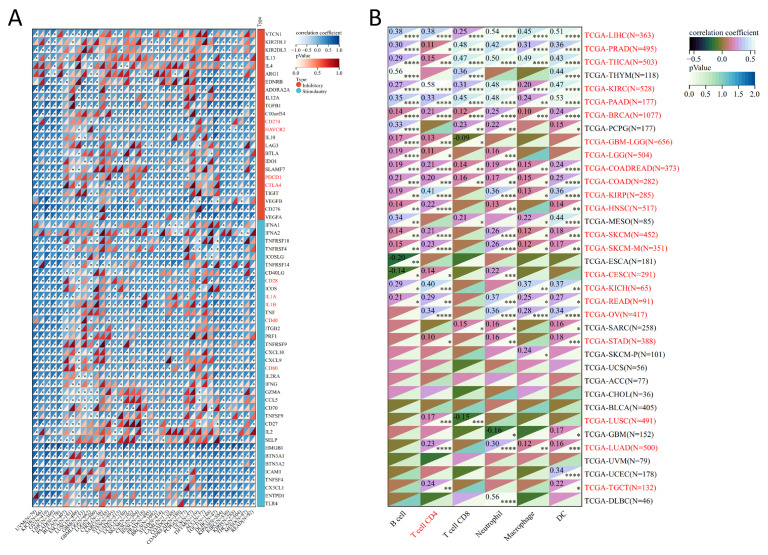
The correlation between immune genes, immune cells and pan−cancer from SangerBox online website (http://sangerbox.com/, accessed on 29 September 2022). (**A**) Relevance of pan−cancer to immune−stimulatory and immune−inhibitory genes. (**B**) The correlation between pan−cancer and immune cells, the results indicate a strong correlation between LIHC and immune cells. * *p* < 0.05, ** *p* < 0.01, *** *p* < 0.001, **** *p* < 0.0001.

**Figure 15 biomolecules-13-00883-f015:**
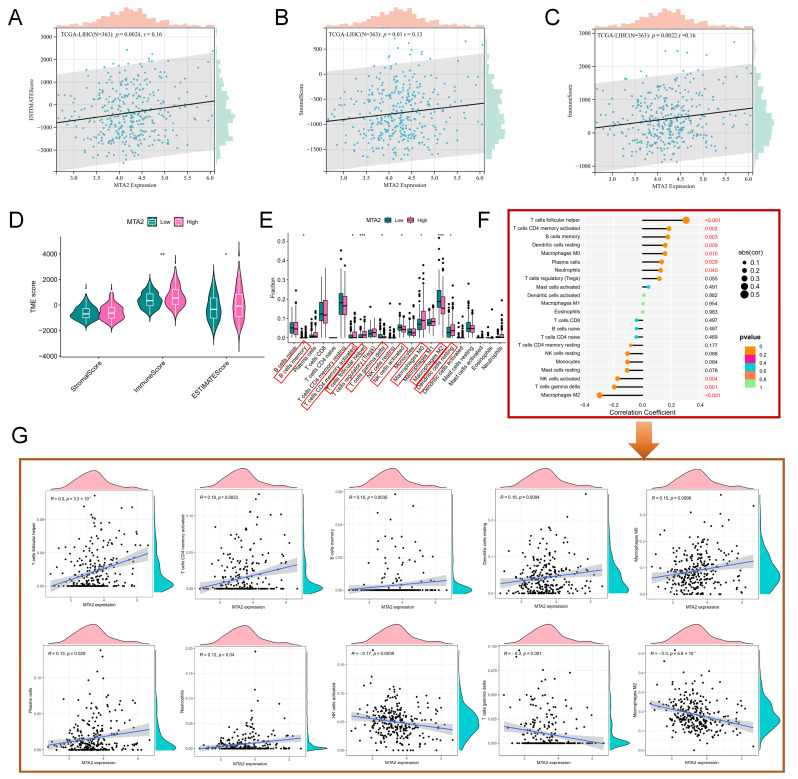
Correlation of MTA2 with the tumor immune microenvironment and immune cells in HCC. (**A**) Correlation of MTA2 with ESTIMATEScore (r = 0.16, *p* = 0.0021). (**B**) Correlation of MTA2 with StromalScore (r = 0.13, *p* = 0.01). (**C**) Correlation of MTA2 with ImmuneScore (r = 0.16, *p* = 0.0022). (**D**) The TME score between high−MTA2 and low−MTA2 groups. Results indicated that the high−MTA2 group was higher than low−MTA2 group in ImmuneScore and ESTIMATEScore. (**E**) The difference in the fraction of immune cells between high−MTA2 and low−MTA2 groups, the box plot showed a higher immune fraction in the high−MTA2 group. (**F**) The relevance of MTA2 to immune cells in HCC. (**G**) T follicular helper cells (r = 0.3, *p* = 3.2 × 10^−7^), CD4 memory activated T cells (r = 0.18, *p* = 0.0023), B cell memory (r = 0.18, *p* = 0.0035), Dendritic cells resting (r = 0.16, *p* = 0.0094), M0 macrophages (r = 0.15, *p* = 0.0098), Plasma cells (r = 0.13, *p* = 0.029), Neutrophils (r = 0.12, *p* = 0.04), NK cells activated (r = −0.17, *p* = 0.0039), T cell gamma delta (r = −0.2, *p* = 0.001), M2 macrophages (r = −0.3, *p* = 5.6 × 10^−7^). * *p* < 0.05, ** *p* < 0.01, *** *p* < 0.001.

**Figure 16 biomolecules-13-00883-f016:**
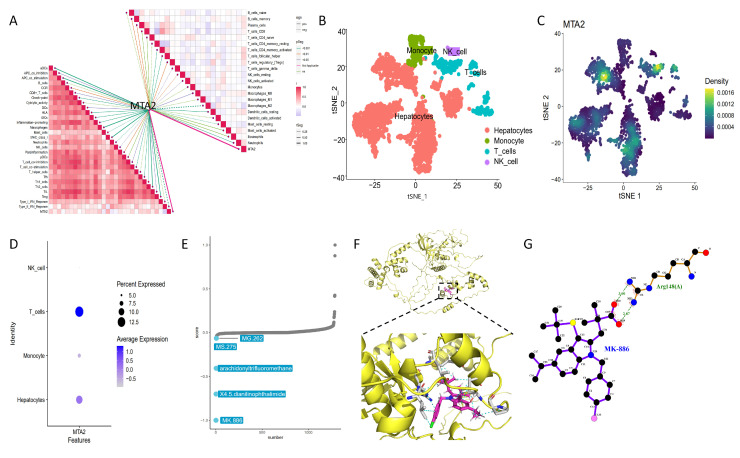
Single cell sequencing analysis and drug screening for HCC. (**A**) Association of MTA2 with immune−related functions and immune cells based on CIBERSORT and ssGSEA results in HCC. According to scRNA−seq analysis, t-SNE plot showed the color−coded clustering of HCC (**B**) and density map (**C**) from GSE146115. (**D**) The bubble map reveals more T cell infiltration in the HCC tissue. (**E**) Drug screening for HCC patients using the Xsum algorithm. The results suggest that MK−886 is the most likely drug to treat HCC. (**F**) Diagram of the molecular docking model between MTA2 and MK−886. The yellow band indicates the MTA2 protein, pink is the small molecule ligand MK−886, grey is the residue of MTA2, blue is the nitrogen atom, green is the chlorine atom and red is the oxygen atom. (**G**) The two-dimensional plane structure model of the interaction between MTA2 protein residues and MK−886.

**Figure 17 biomolecules-13-00883-f017:**
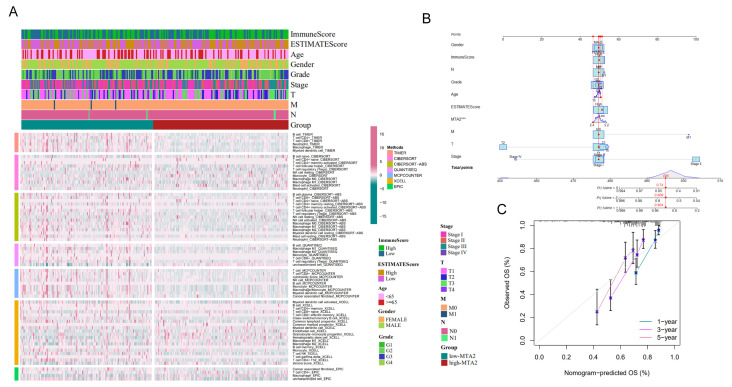
Immune infiltration of the high-MTA2 and low−MTA2 groups in HCC and construction of a nomogram. (**A**) Application of seven bioinformatics algorithms (TIMER, CIBERSORT, CIBERSORT−ABS, QUANTISEQ, MCPCOUNTER, XCELL and EPIC) to analyze the differences in immune infiltration between high−MTA2 and low−MTA2 in HCC. (**B**) Combining the expression of MTA2 and related clinical information (gender, ImmuneScore, grade, age, ESTIMATEScore, T stage, N stage, M stage, and AJCC stage) to construct a nomogram. (**C**) The reliability and accuracy of the nomogram are illustrated by the calibration curves at 1, 3 and 5 years. *** *p* < 0.001.

**Figure 18 biomolecules-13-00883-f018:**
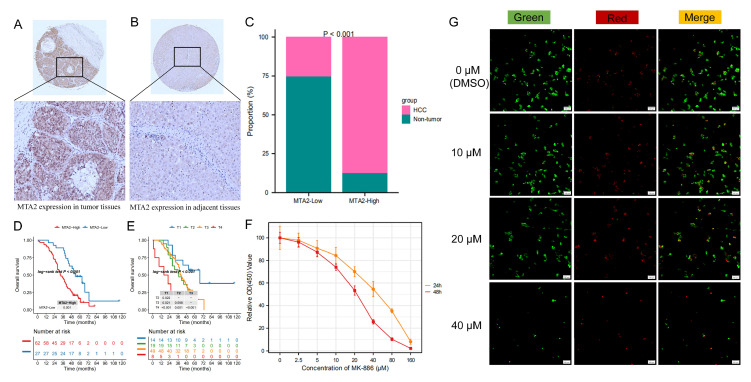
Immunohistochemical assay of MTA2 in HCC tissues and growth inhibition effect of MK−886 on HepG2 cells. (**A**,**B**) Representative immunohistochemical images showed high MTA2 protein expression in HCC tissue and low MTA2 protein expression in paired adjacent nontumorous tissue. (**C**) Differences in MTA2 expression between HCC and normal tissues (*p* < 0.001). (**D**) Comparison of patient survival between high− and low−MTA2 subgroups in HCC (*p* < 0.001). (**E**) In HCC patients, different survival outcomes were revealed when comparing different T stages (*p* < 0.001). (**F**) Growth inhibition effects of MK−886 on cell proliferation detected by CCK−8 array. (**G**) The HepG2 cell viability after treatment with MK−886 for 48 h (green fluorescence represents live cells, red fluorescence represents dead cells).

**Table 1 biomolecules-13-00883-t001:** The AUC values in pan-cancer. Our MTA2 tumor expression data from the TCGA database and normal tissue expression data from the TCGA and GTEx databases.

Cancer	CI	AUC
ACC (TCGA)	0.555~0.731	0.643
BLCA (TCGA)	0.620~0.833	0.727
BRCA (TCGA)	0.687~0.763	0.725
CESC (TCGA)	0.766~0.909	0.838
CHOL (TCGA)	1.000~1.000	1.000
COAD (TCGA)	0.821~0.896	0.859
DLBC (TCGA+GTEx)	0.810~0.892	0.851
ESAD (TCGA+GTEx)	0.682~1.000	0.843
ESCA (TCGA)	0.639~0.984	0.811
GBM (TCGA+GTEx)	0.785~0.845	0.815
HNSC (TCGA)	0.666~0.808	0.737
KICH (TCGA)	0.586~0.809	0.697
KIRC (TCGA)	0.662~0.779	0.720
KIRP (TCGA)	0.525~0.701	0.613
LAML (TCGA+GTEx)	0.426~0.567	0.496
LGG (TCGA+GTEx)	0.854~0.887	0.871
LIHC (TCGA)	0.812~0.905	0.859
LUAD (TCGA)	0.690~0.787	0.739
LUSC (TCGA)	0.907~0.954	0.930
OV (TCGA+GTEx)	0.646~0.747	0.697
PAAD (TCGA+GTEx)	0.954~0.991	0.973
PRAD (TCGA)	0.528~0.692	0.610
READ (TCGA)	0.642~0.908	0.775
SKCM (TCGA+GTEx)	0.617~0.681	0.649
STAD (TCGA)	0.745~0.890	0.818
TGCT (TCGA+GTEx)	0.752~0.857	0.804
THCA (TCGA)	0.627~0.750	0.688
THYM (TCGA+GTEx)	0.885~0.932	0.909
UCEC (TCGA)	0.749~0.850	0.800
USC (TCGA+GTEx)	0.504~0.711	0.608

**Table 2 biomolecules-13-00883-t002:** The association of clinicopathological characteristics with MTA2 expression in 89 patients with HCC.

Characteristics	Number of Patients (n = 89)	MTA2 Expression		*p*-Value
		Low (n = 27)	High (n = 62)	
Age				
<50	32 (36.0%)	12 (13.5%)	20 (22.5%)	
≥50	57 (64.0%)	15 (16.9%)	42 (47.2%)	0.389
Gender				
Female	11 (12.4%)	1 (1.1%)	10 (11.2%)	
Male	78 (87.6%)	26 (29.2%)	52 (58.4%)	0.198
Clinical stage				
I + II	33 (37.1%)	14 (15.7%)	19 (21.4%)	
III + IV	56 (62.9%)	13 (14.6%)	43 (48.3%)	0.096
Tumor				
T1 + T2	33 (37.1%)	14 (15.7%)	19 (21.4%)	
T3 + T4	56 (62.9%)	13 (14.6%)	43 (48.3%)	0.096

## Data Availability

The data underlying this article will be shared on reasonable request to the corresponding author.

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
