# Peer review of "Prognostic, Immunological, and Mutational Analysis of MTA2 in Pan-Cancer and Drug Screening for Hepatocellular Carcinoma"

_biomolecules, 2023, doi:10.3390/biom13060883_

Round 1

Reviewer 1 Report

This is an impressive study for its magnitude scale of data collection. It provides novel and meaningful data. But I have a few suggestion to improve it.

In figure 5, The diagram is a bit cluttered. It would be easier to understand if there was one Kaplan-Meyer plot for each DFS, DDs, and PFS, or if it was presented in a more organized manner.

In figure 7, Similar to the 5th figure, it's difficult to understand due to so many images. Also, it may be very challenging for someone who is unfamiliar with AUC-ROC. Therefore, it would be better to use a simpler diagram that is easy to explain. The correlation between MTA2 and LIHC explained in result 3,4 is difficult to assess in fig. 9B. It would be easier to understand if the degree of correlation was represented with a numerical value below the figure.

The main focus of the paper is on the impact of MTA2 overexpression and its application in ICI. However, mentining p53 and abruptly including fig10 A,B,C may give the impression of being out of context.

Considering the flow of the paper, the intention is to use MTA2 as a marker to determine whether to apply ICI or not. However, the difference shown in fig 12E seems to be too insignificant.

It wound be beneficial if only the gene expressions related to the immune system and ICI were observed in fig 13. So far, there have been too many figures presented, making it difficult to comprehend the information. Alternatively, it may be useful to highlight the genes of particular interest by bolding them.

Request experiment for revision

It the actual cellular level effects were observed by treating the HCC cell lines with MK-866 through in vitro experiments, it would increase the reliability of the statistical values.

Reviewer 2 Report

The manuscript by Huang et al. presents a thorough analysis of the role of MTA2 in cancer generally and accordingly in hepatocellular carcinoma. The authors have performed extensive bioinformatic analysis and have come up with some interesting findings, however some points need to be taken into consideration before publication. 

Here are some specific points to be processed:

Major:

1. In total, the manuscript has a lot of information in figures, but on the contrary, there is actually too little text in between. The authors should provide more explanation in the text, regarding the results in the figures and not just list them in paragraphs.

2. Results: Regarding Figure 2C, the authors state in the Results section that “The box plot illustrated the significant association of MTA2 expression with the development of tumor stage.” This in not a general trend, since statistical significance exists only for 4 cancer types, and even in that case, increased expression of MTA2 is seen in earlier stages e.g., in BLCA, MESO and even PAAD and SKCM. These results should be commented on.

3. Regarding the association between MTA2 expression levels and clinical outcomes of cancer patients, no uniformity exists, since it is evidenced that MTA2 overexpression is linked to either good or bad prognosis depending on the cancer type. This is a controversial finding that potentially weakens the putative pan-cancer prognostic utility of MTA2 and should be explained. 

4. The authors should explain why they chose LIHC to further investigate the role of MTA2. They state that “Because of the high AUC values of MTA2 in TCGA-LIHC and its remarkable correlation with survival in TCGA-LIHC, we performed a series of analyses of MTA2 in TCGA-LIHC”. As they show in Figures 4 and 5, overexpression of MTA2 is strongly associated with poor survival in other types of cancer and not as much in LIHC. In fact, in Figure 2B it is shown that LIHC is the cancer type with the least overexpression of MTA2, compared with all other cancer types. Thus, the authors should provide a rationale for choosing LIHC, based only on tissue database data.

5. Using HCC tissues, the authors showed that overexpression of MTA2 was correlated with poor survival, which is more evident than the TCGA data. The authors could correlate also T stage with survival to show if there is any significant correlation based on staging.

Minor: 

1. Abbreviations should be explained in the main text separately from the abstract, e.g. line 43 HCC.

2. Introduction: Too much unlinked information, please divide it into more paragraphs. 

3. The paragraph describing the scope of the study needs to be more precise and targeted. All the details regarding software and protocols should be included in the Materials and Methods section.

4. Materials and Methods: lines 212-217, if the score scale has been previously used, please add reference.

5. In Kaplan Meier plots, high vs. low expression levels are defined based on median values? Please clarify. 

6. Figures 6E and 6F have no connection with the rest of the panels of Figure 6 and should be separate.

Round 2

Reviewer 2 Report

The authors have sufficiently resolved all of the suggestions and comments.

I have no further comments.